# Critical Admission Temperature of H_2_ and CH_4_ in Nanopores of Exchanged ERI Zeolites

**DOI:** 10.3390/nano9020160

**Published:** 2019-01-29

**Authors:** Karla Quiroz-Estrada, Miguel Ángel Hernández, Carlos Felipe-Mendoza, Juana Deisy Santamaría-Juárez, Vitalii Petranovskii, Efraín Rubio

**Affiliations:** 1Doctorado en Nanociencias y Micro-Nanotecnologías, UPIBI, Instituto Politécnico Nacional, Ciudad de México 07340, Mexico; k.quiroz.estrada@gmail.com; 2Departamento de Investigación en Zeolitas, Benemérita Universidad Autónoma de Puebla, Puebla 72570, Mexico; 3Departamento de Biociencias e Ingeniería, CIIEMAD, Instituto Politécnico Nacional, Ciudad de México 07340, Mexico; cfelipe@ipn.mx; 4Facultad de Ingeniería Química, Benemérita Universidad Autónoma de Puebla, Puebla 72570, Mexico; deisy.santamaria@correo.buap.mx; 5Centro de Nanociencias y Nanotecnología, Universidad Nacional Autónoma de México, Carretera Tijuana-Ensenada, Km. 107, Ensenada 22860, Mexico; vitalii@cnyn.unam.mx; 6Centro Universitario de Vinculación y Transferencia de Tecnología, Benemérita Universidad Autónoma de Puebla, Puebla 72570, Mexico; efrainrubio@yahoo.com

**Keywords:** hydrogen, methane, greenhouse gas, erionite

## Abstract

Due to the nanoporous nature of zeolitic materials, they can be used as gas adsorbents. This paper describes the effect of critical admission temperature through narrow pores of natural ERI zeolites at low levels of coverage. This phenomenon occurs by adsorption of CH_4_ and H_2_ on pores in natural erionite. The zeolite was exchanged with aqueous solutions of Na^+^, Mg^2+^, and Ca^2+^ salts at different concentrations, times, and temperatures of treatment. Experimental data of CH_4_ and H_2_ adsorption were treated by the Langmuir equation. Complementarily, the degree of interaction of these gases with these zeolites was evaluated by the evolution of isosteric heats of adsorption. The Ca^2+^ and Mg^2+^ cations favor the adsorption phenomena of H_2_ and CH_4_. These cations occupy sites in strategic positions Ca1, Ca2, and Ca3, which are located in the nanocavities of erionite zeolites and K2 in the center of 8MR. Following the conditions of temperature and the exchange treatment, ERICa2 and ERINa3 samples showed the best behavior for CH_4_ and H_2_ adsorption.

## 1. Introduction

It has already been established that greenhouse gases contribute significantly to global warming. Due to this circumstance, the capture, treatment, and confinement of these substances became very relevant issues [1]. There is a myriad of materials that can cover these aspects. However, zeolites, MOFs (metal oxide frameworks), silicas, carbon-based materials, and clays are especially important. Nevertheless, the synthesis processes of many of these adsorbents are very complex and time-consuming [2]. It was reported that substantial energy requirements have to be met in order to generate assorted zeolite adsorbents. The development of optimum adsorbents endowed with satisfactory adsorption capacities, fine chemical selectivity, and straightforward synthesis and regeneration, remain a challenge so far. In relation to this subject, the adsorption capturing process has aroused great interest due to the countless advantages that can provide, such as high energy efficiency, easy monitoring, and relatively low capital investment costs [3]. In recent years, the use and preparation of small pore zeolites developed different applications, mainly owing to the fact that they are nanoporous adsorbents which can be used in both catalysis and gas separation [4]. These nanostructures, endowed with small pores, are especially relevant for capturing greenhouse gases from large emission sources. Such nanostructures could be potentially used both to reduce the energy fuels demand for achieving the separation of greenhouse gases, i.e., CH_4_, H_2_, or CO_2_ from flue gas [5], and to find an alternative to the storage of hydrogen and methane, since these gases are a source of clean energies [6,7]. Among the most investigated materials for this propose are synthetic zeolites [8,9,10,11], MOFs, [12] and covalent organic frameworks (COFs) [13,14]. Zeolites are classified by the International Zeolite Association (IZA) according to the size of their channels, which are related with the number of atoms or members (those in coordination with aluminum and silicon oxide tetrahedrons) in the rings. Pores with eight atom rings are classified as “small pore zeolites” followed by openings of 10 atoms as “medium pore zeolites”. After that, zeolites with rings of 12 atoms are designated as “large pore zeolites” (12-rings), and finally, those with more than 12 member rings are labeled as “extra-large pore zeolites” [15,16]. Another way to classify zeolites is according to the number of pores direction. Zeolites with channels along one, two, or three directions are denominated as “one-dimensional”, “two-dimensional” or “three-dimensional”, respectively [17]. Between the most abundant Mexican natural zeolites, erionite (ERI) represents a three-dimensional small pore structure, with a pore architecture of 8 × 8 × 8 3D channel system [1]. For this purpose, eight-ring zeolites are of great interest since the effective kinetic diameters displayed by greenhouse gases are smaller than the N_2_ molecule within porous solids, in contrast to their diameters in the gaseous state [18]. It is expected that small-pore zeolites will exert an increased selectivity towards greenhouse gases [19]. Generally, there are two types of porosity in natural zeolites: (i) A primary one, pertinent to the presence of micropores, and (ii) a secondary one, due to mesopores. The secondary porosity allows such phenomena as adsorption of large molecules [20]. In contrast, the primary porosity is characterized by a micropore volume (W_0_), in which volumetric filling, rather than layer-to-layer adsorption takes place [21]. Several conventional methods exist to characterize the microporosity of zeolites. In order to evaluate the adsorption of gas molecules, models such as those of Langmuir, Dubinin-Astakhov, and Differential Curves of Adsorption are employed and widely used in high-pressure adsorption processes [22,23]. The ERI framework type is a small-pore zeolite with an aperture of eight member rings (8MR), a hexagonal unit cell with parameters: a = b = 1:33 nm; c = 1:51 nm, α = β = 90°; γ = 120°. The chemical formula of a typical erionite material is (Ca, Mg, Na_2_, K_2_)_4.5_ [Al_9_Si_27_O_72_]·27 H_2_O, with a Si:Al molar relation (MR) equal to 3. This crystalline structure is characterized by a three-dimensional channel system, along and perpendicular to the [001] direction. Erionite consists of three structural units: (i) Hexagonal prisms, (ii) cancrinite supercages (ε units), and (iii) erionite cavities, with a pore opening of 0.36 nm × 0.51 nm (Figure 1) [24]. All types of natural zeolites are polycationic. In the case of erionite, Figure 1 displays five types of cationic sites Ca1, Ca2, Ca3, K1, and K2, where K1 is found in the internal framework of cancrinite cages and K2 is at 8MR entry (Figure 1) [25]. 

## 2. Materials and Methods

The erionite zeolites used in this work were collected in Agua Prieta, Sonora, Mexico. These zeolites were passed through a 60–80 mesh size and chemically treated with sodium, calcium and magnesium salts 0.01N (J.T.Baker, Phillipsburg, NJ, USA), each one at three different exchange cycles. After that, zeolites were washed again with distilled water in order to eliminate Cl^−^, which was confirmed with AgNO_3_ as indicator and finally dried at room temperature. The obtained materials were labeled according to the ERI prefix, exchange cation, and treatment number. 

### 2.1. Characterization

Characterization of natural and chemically modified zeolites by N_2_ adsorption at 77 K, X-ray powder diffraction (XRPD), and energy dispersive spectrometry (EDS) were reported in previous works [1,26]. Additionally, to understand the behavior of natural erionite under thermal treatment, TGA and DSC analysis were carried out. Furthermore, high-resolution scanning electron microscopy (HRSEM) was used to visualize the erionites morphology, and the elemental chemical analysis (wt %) was determined by FRX. Finally, the pore size distribution of the samples was calculated by Dubinin-Astakhov equation and adsorption differential curves from N_2_ isotherms.

XDPR pattern of natural zeolite was determined through a Bruker D8 diffractometer (Bruker, Co., Billerica, MA, USA) using nickel-filtered Cu Kα (λ = 0.154 nm) radiation operated at 40 kV and 30 mA. The patron was refined by the Rietveld method to confirm the composition of crystalline phases. Photomicrographs of the samples under study were obtained with a JEOL, model JSM-7800F (JEOL USA, Inc., Peabody, MA, USA) high-resolution scanning electron microscope at 5 kV. The samples were mounted on aluminium stub holders and subsequently coated with Au using a sputtering coater. Quantitative chemical analyzes were obtained through QUANT-EXPRESS method (Fundamental Parameters) in the sodium (Na) to Uranium (U) range in an X-ray fluorescence spectrometer (S8-TIGER, Bruker, Co., Billerica, MA, USA). Thermogravimetric analysis (TGA) and differential scanning calorimetry (DSC) results of the erionite samples were obtained in a simultaneous thermal analysis equipment NETZSCH STA 449F3 (Erich NETZSCH GmbH & Co., Selb, Germany) with heating ramp of 10.0 °C/min using a N_2_ flow of 10 mL min^−1^ in a temperature range of 25 to 800 °C.

### 2.2. Adsorption of H_2_ and CH_4_

The adsorption experiments were carried out using the gas chromatography technique, where the interaction of the CH_4_ and H_2_ molecules was evaluated at low coverage levels. This allowed us to evaluate the enrichment of the surface of zeolites with gas molecules without taking into account the lateral interactions of the adsorbed gas. Erionite samples packed inside of chromatographic columns (internal diameter = 5 mm; length = 50 cm) were pretreated *in situ* under a flow of the carrier gas at 573 K before the adsorptive injection. 

Some important physical properties of the adsorptives employed in this work are listed in Table 1. Note that kinetic diameters (σ, nm) of the adsorptive molecules [27] are comparable to the sizes of the pore openings delimited by eight atoms in the erionite crystal structure [28,29]. 

The adsorption chromatograms were obtained in a GOW-MAC gas chromatograph (GOW-MAC Instrument Co., Bethlehem, PA, USA) equipped with a thermal conductivity detector at five different temperatures: 298, 353, 393, 433, and 473 K. Retention times were measured using helium (30 cm^3^ min^−1^) as carrier gas. The obtained data were evaluated through the GC peak maxima method [30] in order to obtain the adsorption isotherms, and finally, the energetic interaction was analyzed by the behavior of isosteric heat of adsorption (qst), which was calculated by applying the Clausius Clapeyron equation:(1)$${\left[\frac{\partial \ln p}{\partial T}\right]}_{a}=\frac{q_{st}\left(a\right)}{RT^{2}}$$

This value, also called isostatic enthalpy of adsorption, represents the energy of interaction between the zeolite and the adsorbed gases. This equation assumes the adsorbate behaves as an ideal gas and the molar volume of the adsorbed phase is minimal compared to that of the adsorbate [31].

Once the experimental adsorption isotherms were obtained, the Langmuir mathematical model was evaluated from its equation in linear form (Equation (2). Adsorption parameters were calculated through linear regression.
(2)$$\frac{1}{a}=\frac{1}{a_{m}}+\left(\frac{1}{a_{m}K_{P}}\cdot \frac{1}{P}\right)$$
where *a_m_* correspond to the monolayer adsorption capacity.

## 3. Results

### 3.1. Characterization

Figure 2 shows the XRPD pattern of the natural sample, where the presence of the crystalline phase corresponding to an erionite zeolite is confirmed with characteristic signals in the diffraction angles 2θ: 7.7 °, 11.76 °, 13.34 °, 19.44 °, 21.3 °, 23.48 °, 25.17 °, 27.05 °, 28 °, 31.42 °, 33.41 °, and 35.89°, accompanied by another type of eight member rings zeolite (chabazite) and quartz. The pattern was evaluated using the High Score Plus 3.0 software, where semiquantitative percentages of the phases were determined by applying Rietveld’s refinement (Goodness of fit = 0.69; R expected = 14; R profile = 12), which gave us a good estimate of these percentages. In this sample, the main percentage obtained corresponds to erionite crystalline phase with 75.1% (blue), while chabazite is equal to 24.1% (green), and finally, minor amounts of quartz are equal to <1%. The patterns of exchange erionites were studied previously [26], and it is observed that there are no significant differences between the obtained patterns.

HRSEM micrographs (Figure 3) clearly reveal that the nanostructures of the sample correspond to natural erionite (Figure 3a–c). The erionite crystalline structure occurs as bundles of parallel fibrils [32] with a diameter between 25 and 31 nm and variable lengths. Also, it can be observed that crystals contain some impurities because they are natural samples.

In Table 2, the elemental chemical composition determined by FRX in wt % is observed. First, the Si/Al ratio does not show a significant variation between exchanged erionites. 

In the sample of natural eronite without treatment, there are five main sites where the cations are located according to the deficiency of charges of the structure and its coordination with the water molecules. According to the obtained data, it can be assumed that there is a reordering of cations in the different preferential sites. First, in the sodium exchanged samples, Na^+^ increases as the treatment cycle increases, while the percentage of K^+^ decreases considerably compared to the natural one. The preferential sites for sodium are reported as Ca2 and Ca3 [33], and when performing the exchange treatment with NaCl, the K cations are replaced with Na^+^ in K2. In the case of calcium, its original position is the Ca2 site [34], and when performing the treatments with CaCl_2_, the amount of Mg^2+^, K^+^ and Na^+^ decreases in the Ca1, K2, and Ca3 sites, with the increase of Ca^2+^ in the same. Finally, Mg saturation occurs mainly with the replacement of Na^+^ at Ca3 and K^+^ in K2, in addition to their preferential site Ca1 [35]. This does not take into account the exchange in the preferential sites of chabazite as a secondary crystalline phase. On the other hand, this table also shows a considerable Fe_2_O_3_ content and minor amounts of TiO_2_. These compounds have been reported in other Erionite zeolites as accompanying phases and are attributed to heterogeneous and amorphous oxides that are not visible by XRPD [34,35,36]. The chemical variability of erionite may be associated with the percentage of chabazite present (Figure 2) due to exchange treatments act on the cationic sites of both chabazite and erionite. 

The t-plots were evaluated from isotherms of nitrogen adsorption at 77 K for the natural and exchanged erionite zeolites. Figure 4a shows that the statistical thickness of the multilayer adsorbed in nm (t) varies between 0.14–1.0 nm. The deviations represent the existence and dimension of the pores in the solid [37]. This plot exhibited three characteristic features of a linear behavior explained below: Initially, in the range of 0.16 nm < t < 0.2 nm represents the progressive growth of the thickness t of the adsorbed layer in micropores, followed by capillary condensation process in the mesopores of the sample (0.2 nm < t < 0.35 nm), and finally, the adsorption in the external surface area (0.35 nm < t < 1 nm), which is observed in the third stage. The empirical method, proposed by Zhu and Lu [23] names differential adsorption curves (DAC), representing the variations of the volume adsorbed with respect t. Figure 4b shows the nanopore size distribution obtained by this method, where the main signals were founded in 0.56 nm. These values correspond to the open cavities of eight members rings of erionite zeolites.

Another method to evaluate the PSD is from Dubinin-Astakhov (DA) equation (Table 3). In this table, ERIN shows a value of 0.51 nm. Meanwhile, the samples exchanged with Na^+^ had a decrease of PSD due to fact Na^+^ was positioned in the empty K2 site near to the center of 8MR, which results in the reduction of the available space to the entry to some molecules and the increase of micropore volume. On the other hand, zeolites rings are flexible. In base of chemical compositions, it could be speculated that in the case of Ca^2+^ and Mg^2+^ exchanged erionites, pores are expanded, owing to the cation’s divalent nature. Results suggest the pore diameter increments are due to these cations occupy mainly a Ca2, Ca1, and Ca3, Ca1 sites, respectively. Micropore volume values present a considerable increase. This mainly due to the elimination of some amorphous impurities and the rearrangement and substitution of the cations in the cavities of erionite and chabazite. The atomic radius of Ca^2+^, Mg^2+^ and Na^+^ are 0.194 nm, 0.15 nm, and 0.18 nm, respectively. The fact that Ca^2+^, Mg^2+^ and Na^+^ have values shorter than K^+^ (0.22 nm) leave more available micropore volume to absorb some molecules as H_2_ or CH_4_. This increase is reflected in surface area values [26]. 

TGA and DSC profiles of the erionite samples in the temperature range of 298–1073 K are shown in Figure 5. It can be observed a total weight loss of 13% for the natural sample ERIN. The cation exchange treatments performed on the natural sample show better behaviors against weight loss resistance with the following decreasing sequence: ERICa1 > ERIMg2 > ERINa3 > ERINa2 > ERICa3 > ERICa2 > ERIMg3 > ERINa1 > ERIMg2, in which a continuous water loss behavior between 323 and 453 K is revealed (Figure 5a). The accumulated loss between 50 and 175 °C is related to the loss of water placed in the supercages of the zeolitic structure that corresponds mainly to desorption of physisorbed water [38]. The next section, from 473 K until 873 K, shows the losses related to the molecules more bound to the cavities containing the cations at the entry of said pores, possibly to the strongest ionic polarization capacity for water molecules in this type of zeolites [39]. The loss of water continues at a very low speed up to 873 K, attributed to the dehydration of the chemisorbed water possibly due to the dihydroxylation process. This process originates from the destruction of hydroxyl bonds, which were formed when water molecules were polarized by the exchanged cations, resulting in an evacuation of more hosted water inside of zeolite cavities [40]. On the other hand, Figure 5b represents DSC analysis, which reveals the totality of energies required by processes involved. The volatilization of water includes the rupture of bonds with cations, the crystalline network, and with other molecules of water, in addition to the diffusion of water inside the cavities and the reorganization of the crystal lattice [41]. Erionite samples with wide exothermic signals are observed in the range of 473 to 903 K with a maximum at 623 K for the samples ERINa1, ERINa2, ERICa2, ERICa3, ERIMg1, and ERIMg3, which is the result of the dehydration of the coordination of water molecules with cations that cannot be located in specific fixed positions. The heat flux expressed by the samples ERINa3, ERICa1, and ERIMg2, represents the loss of weakly bound water molecules followed by the desorption of water trapped inside the boxes and the progressive dehydration of the material (water adsorbed physically or chemically) [42]. 

### 3.2. Adsorption Dependence with the Admission Temperature

A series of erionite zeolite with diverse silicon-aluminium composition ratios (Si:Al) and several cation types (Ca^2+^, Mg^2+^, and Na^+^) were obtained. These samples were analyzed experimentally in the H_2_ and CH_4_ adsorption process at a low coverage level by gas chromatography. This technique allows to study the adsorption capacity caused by the interaction between gas and zeolite surface at low or zero coverage at five different temperatures in order to evaluate the qst behavior. Gas diffusion through erionite channels and retention times are responsible of the pressure variability due to the partial blockage of the cavity’s entrances by cations. Figure 6, Figure 7, Figure 8 and Figure 9 present CH_4_ and H_2_ adsorption isotherms on ERI zeolites measured at temperature range of 298 and 473 K. From these figures, it is evident the higher adsorption capacity of both H_2_ and CH_4_ depicted by the ERICa2 and ERINa3 zeolites, while the lower adsorption capacity corresponds to erionites exchanged with Mg with their three treatments for all experimental temperatures.

Figure 6a–d and Figure 7a–f show a better adsorption capacity of CH_4_ at room temperature in contrast with H_2_ adsorption with the samples exchanged with Ca^2+^ and Mg^2+^. In these samples, the highest adsorption values are shown above 433 K (Figure 9a–f). This last temperature is close to the value, where, according to the TGA analysis (Figure 5), the weight loss is associated with the evaporation of water molecules that are in coordination with exchange cations. It was reported that the dehydration of erionite zeolites has a direct relationship with the mobility of cations in the structure. Therefore, the loss of water molecules causes a shift to improve their coordination with the oxygen atoms of the structure which results in the migration of cations to K2 site [43]. 

On the other hand, the Langmuir model is based on the assumption that adsorption is limited by a fixed number of adsorption sites and there is no interaction between the adsorbed molecules [44]. Its most characteristic parameter represents the maximum adsorption capacity in monolayer (am), indicative of high energy interactions between adsorbate and adsorbent. Taking into account the behavior of this parameter regarding temperature, the accessibility of erionite channels (8MR) for the gas molecules adsorption shows a temperature dependence. A total pore opening is caused by electrostatic forces at certain conditions of pressure and temperature, which allow H_2_ and CH_4_ molecules to permeate the pore network. This point was defined as the critical admission temperature (T_C_), or the so-called pore-blockage temperature [45]. As shown in Figure 10, ERICa2, ERINa3, and ERINa1 erionite zeolites with Si:Al = 3.99, 3.7382 and 4.07 respectively, revealed noticeable adsorption of H_2_ and for CH_4_ at temperatures higher than 420 K. This behavior has been observed in narrow pore zeolites. The cations in the ring are agitated by thermal motion due to different factors as: (i) Pore apertures suffer a thermal dilatation, (ii) aperture atoms by “pulsation”, (iii) number of cations per unit cell, and (iv) the diffusion of gas molecules activated by high temperatures [8,46]. However, this phenomenon is not visible for CH_4_ and H_2_ adsorption to the rest of zeolites exchanged, thus indicating the existence of no pore blockage for the sorption of these gases even at temperatures as low as 435 K. 

The effect of the replacement of Mg^2+^ or Ca^2+^ ions by Na^+^ cations causes a decrease in the diameter of the input windows to values of approximately 0.3 nm. The divalent ions in the ERI zeolites are found mainly in the Ca1, Ca2, and K2 sites. The absence of high temperatures results in partially closed rings and a low diffusion of the H_2_ (σ = 0.364 nm) and CH_4_ (σ = 0.38 nm) adsorbed molecules. The opening of 8MR without cation with a diameter of 0.51 nm decrease with the presence of exchange cations with an atomic radius as mentioned in Section 3.1 at the K2 site. For that reason, the explanations related to the pulsation of the opening atoms can be discarded. Therefore, the admission of any molecule in erionite supercage is only possible if the cations located in K2 migrate at least partially from the neck of the 8MR after having been subjected to thermal and structural stimuli.

### 3.3. Isosteric Heat of Adsorption

Figure 11 shows the behavior of qst as a function of the amount adsorbed. The qst values related to the ERI substrate correspond to the following decreasing sequence, qst (CH_4_): Mg: 1 > 2 > 3; Ca: 2 > 1 > 3; Na: 1 > 2 > 3, while for qst (H_2_): Mg: 1 > 2 > 3, Ca: 2 > 3 > 1, and Na: 2 > 3 > 1. First, the qst values proceeding from CH_4_ and H_2_ adsorption on ERICa1, ERICa2 ERIMg1, and ERIMg2 are larger than the enthalpies of vaporization of adsorbate gas, which means CH_4_ and H_2_ molecules have a strong interaction with erionite surfaces, then a cohesive interaction between the same gas molecules takes place. In these erionite zeolites, the replacement of Mg^2+^ or Ca^2+^ ions at the Ca1, Ca2, and Ca3 positions cause an enlargement of the free space of the input windows thus increasing the value of the effective diameter of the input windows and improving the interaction energy, which is related mainly with two factors. First, the cations position in each adsorbent allows to act as a molecular sieve, and second, the hydrogen atoms have an attraction to oxygen atoms of zeolites, which cause different qsts for each zeolite. Qst is dependently mainly of the surface. In the case of physisorption, Van der Waals effects between CH_4_-CH_4_ and H_2_-H_2_ molecules are related with the decrement of qst value with the increase of the amount adsorbed in a heterogeneous surface energetically [47]. In the case of the natural and exchanged with sodium Na^+^, ERICa3, and ERIMg3 samples, the behavior of qst can be associated with weak interactions on a nearly uniform energetic surface. 

## 4. Conclusions

The crystalline structures were essentially displayed as elongated prisms of erionite fibers with diameters in the range of 25–30 nm. Treatments with Na^+^, Ca^2+^, and Mg^2+^ cations in the natural erionites allowed the opening and blocking of pore entrances by the effect of critical admission temperature. According to the results obtained, for the selection of a zeolitic adsorbent in the capture of CH_4_ and H_2_, the following criteria should be taken into account: (i) Cation saturation in the cavities, (ii) adsorption capacity, (iii) adsorption Tc, and (iv) energetic interactions (qst).

Therefore, for this research, the replacement with the calcium ions in Ca2, Ca3 and K2 sites could be an alternative to the capture or storage of H_2_ and CH_4_. The most fitting treatment was the one with two cycles of CaCl_2_ exchange (ERICa2), that showed the largest opening of the pore diameter according to the DA method. This sample also showed the greatest capacity of H_2_ and CH_4_ adsorption evaluated by gas chromatography at Tc = 420 K. In addition, this material presented a qst behavior corresponding to strong energetic interactions in a heterogeneous surface. 

## Figures and Tables

**Figure 1 nanomaterials-09-00160-f001:**
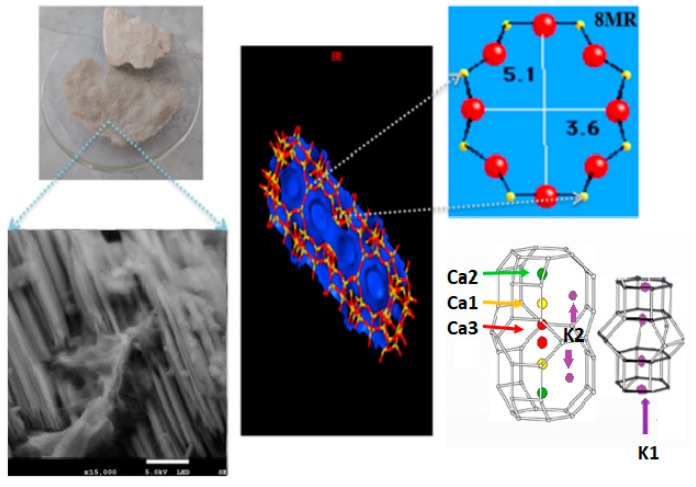
Erionite structure, International Zeolite Association (IZA).

**Figure 2 nanomaterials-09-00160-f002:**
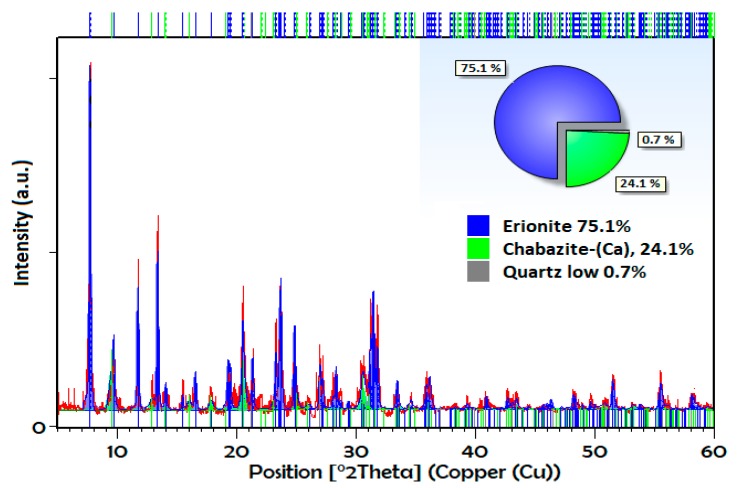
X-ray powder diffraction pattern of natural erionite refined by Rietveld method.

**Figure 3 nanomaterials-09-00160-f003:**
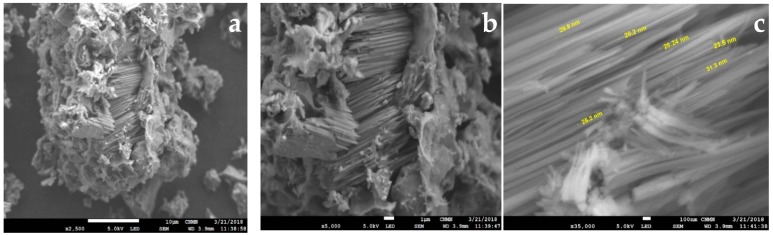
SEM micrographs of zeolite fibers: (**a**) ERIN—×2500, (**b**) ERIN—×5000, (**c**) ERIN—×50,000.

**Figure 4 nanomaterials-09-00160-f004:**
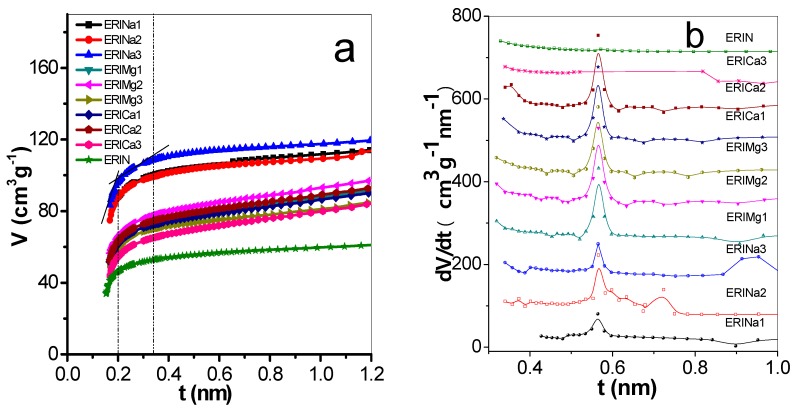
(**a**) t-plots and (**b**) pore size distribution by differential curves of adsorption of erionite zeolites.

**Figure 5 nanomaterials-09-00160-f005:**
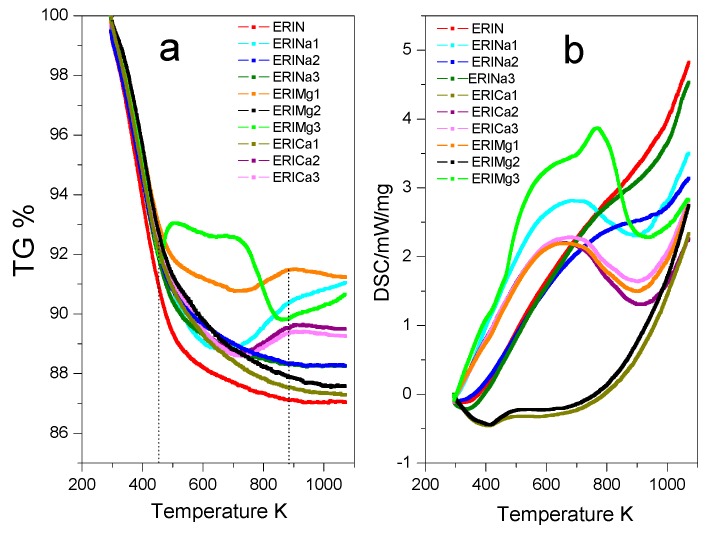
(**a**) Thermogravimetric analysis (TGA) and (**b**) Differential scanning calorimetry (DSC) of erionite zeolites.

**Figure 6 nanomaterials-09-00160-f006:**
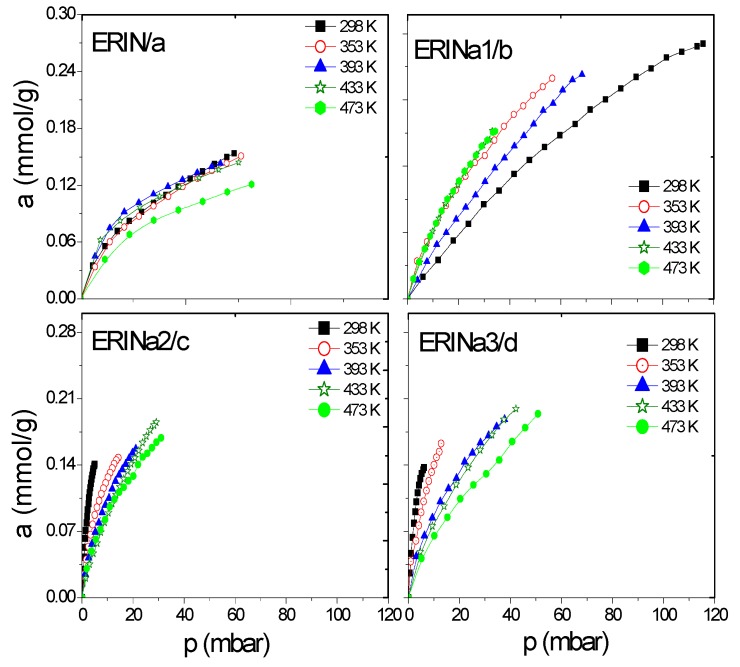
CH_4_ adsorption isotherms at different temperatures on: (**a**) ERIN, (**b**–**d**) Na1–3 exchanged zeolites.

**Figure 7 nanomaterials-09-00160-f007:**
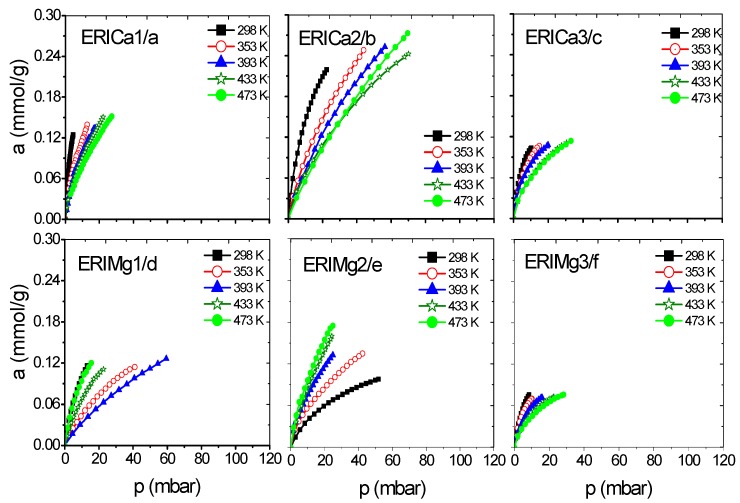
CH_4_ adsorption isotherms at different temperatures on: (**a**–**c**) ERICa and (**d**–**f**) ERIMg exchanged zeolites.

**Figure 8 nanomaterials-09-00160-f008:**
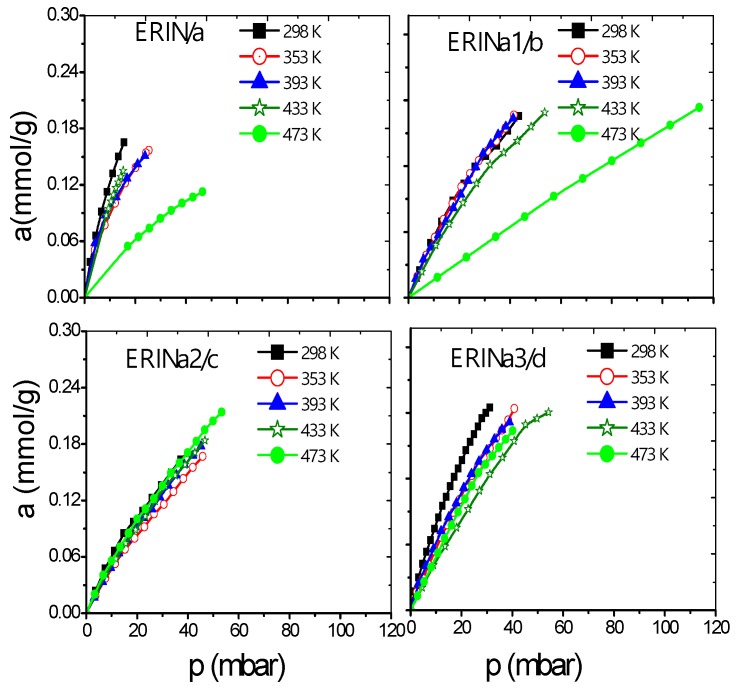
H_2_ adsorption isotherms at different temperatures on: (**a**) ERIN, (**b**–**d**) ERINa1–3 exchanged zeolites.

**Figure 9 nanomaterials-09-00160-f009:**
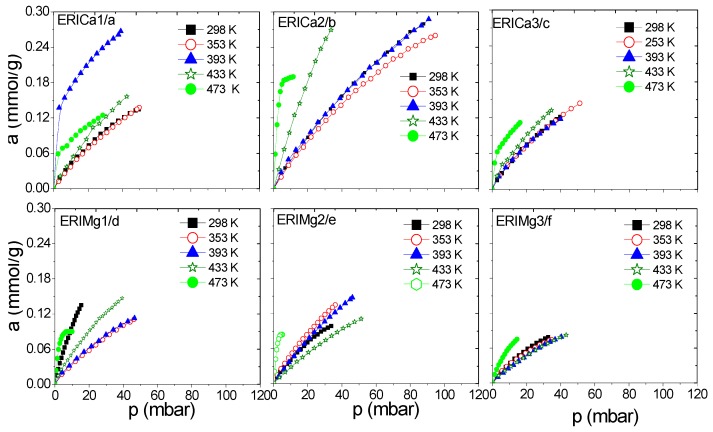
H_2_ adsorption isotherms at different temperatures on: (**a**–**c**) ERICa, and (**d**–**f**) ERIMg exchanged zeolites.

**Figure 10 nanomaterials-09-00160-f010:**
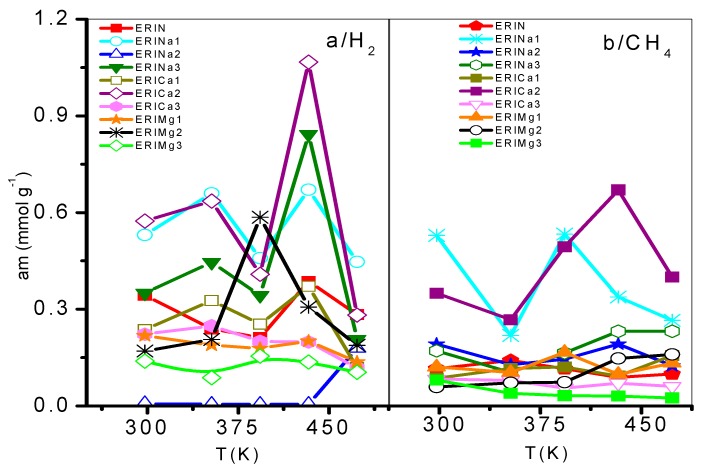
Saturation adsorption limit of: (**a**) H_2_ and (**b**) CH_4_ on ion exchanged erionites.

**Figure 11 nanomaterials-09-00160-f011:**
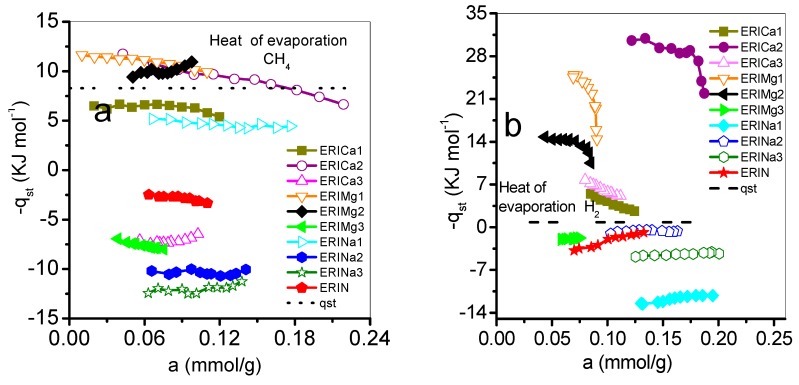
Variation of the isosteric heat of adsorption (-qst) with adsorbate coverage on erionite zeolites: (**a**) CH_4_, (**b**) H_2_.

**Table 1 nanomaterials-09-00160-t001:** Molecular properties of adsorbed gases.

Gas	Polarizability Å^3^	Quadrupole Moment, D Å	Kinetic Diameter, nm
H_2_	0.79	0.52	0.29
CH_4_	2.45	0	0.38

**Table 2 nanomaterials-09-00160-t002:** Chemical composition of erionite zeolites by FRX (wt %).

	ERINat	ERINa1	ERINa2	ERINa3	ERIC1	ERICa2	ERICa3	ERIMg1	ERIMg2	ERIMg3
**SiO_2_**	59.4	59.80	60.90	60.90	64.30	62.30	60.20	61.20	60.80	59.20
**Al_2_O_3_**	12.9	13.50	13.30	14.30	13.60	13.80	13.30	13.20	12.80	13.00
**Na_2_O**	3.52	3.76	4.69	3.85	2.80	2.40	1.21	3.07	2.55	2.94
**K_2_O**	3.56	3.20	2.66	3.35	2.66	3.07	3.13	3.21	3.17	2.76
**Fe_2_O_3_**	2.29	2.23	1.71	2.39	1.81	1.74	2.05	1.83	1.81	1.89
**MgO**	1.39	1.45	1.45	1.47	1.13	1.12	1.32	1.75	1.70	1.96
**CaO**	1.31	1.12	1.46	1.48	2.24	2.72	3.44	1.29	1.33	1.35
**TiO_2_**	0.4	0.19	0.20	0.23	0.20	0.20	0.32	-	0.18	0.24
**H_2_O**	15.23	14.75	13.63	12.03	11.26	12.65	15.03	14.45	15.66	16.66
**Total**	100	100	100	100	100	100	100	100	100	100
**Si/Al**	3.91	4.07	4.04	3.76	4.18	3.99	4.00	4.09	4.20	4.02

**Table 3 nanomaterials-09-00160-t003:** Micropore volume (cm^3^/g), pore diameter (nm) (Dubinin-Astakhov theory—DA) and surface area (m^2^/g) (Langmuir method) of erionite zeolites.

Sample	Wo cm^3^/g	Dp nm	As_L_ [26] m^2^/g
ERIN	0.09	0.51	243.9
ERINa1	0.17	0.49	244.7
ERINa2	0.17	0.48	504.4
ERINa3	0.18	0.47	545.7
ERIMg1	0.13	0.54	322.7
ERIMg2	0.14	0.54	343.8
ERIMg3	0.13	0.53	309.4
ERICa1	0.13	0.55	325.5
ERICa2	0.14	0.54	331.5
ERICa3	0.12	0.56	293.8

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
