# Peer review of "Critical Admission Temperature of H2 and CH4 in Nanopores of Exchanged ERI Zeolites"

_nanomaterials, 2019, doi:10.3390/nano9020160_

Round 1
Reviewer 1 Report
General comments.
The development of new efficient greenhouse gases separation techniques in order to improve current expensive and high-energy-demanding industrial separation processes is a matter of increasing interest. Among the different materials proposed, zeolites offer unique properties for their application in this area due to the combination of regular pore architecture, large adsorption capacity, and high structural stability. Moreover, the use of natural zeolites collected locally has the additional value of sustainability. Thus, the work is of interest, and the results presented as well. However, part of the description of the sample preparation and characterization was described in a previous paper and not included here, and this makes it very difficult to correlated physico-chemical properties (composition for instance) with adsorption properties and it is hard for the reader to follow the discussion.
Moreover, in my opinion, the authors should give a more detailed explanation of the adsorption experiments. This would allow readers not familiarized with the application of gas adsorption to understand the paper better.
The manuscript should be modified before its publication in Nanomaterials.
More details are given below.
Specific comments.
1. English usage should be revised
2. Introduction: small pore zeolites have been studied for separation of light hydrocarbons (olefins from paraffins), for removal of CO2 from natural gas (CO2-CH4 separation) or for CO2 capture to reduce the atmospheric CO2 concentration. The authors study methane and H2. Please justify this according to a possible application.
3. Introduction: some references could be included:
a. A recent revision on small pore zeolites with application in gas separation: Moliner et al., Chem. Mater. 2014, 26, 246−258
b. Shang et al., J. Am. Chem. Soc. 2012, 134, 19246. Use of cation exchanged (Na, K, Cs) CHA for CO2/CH4 separations. The paper shows how the different ability of gas molecules to induce cation deviation from their 8-ring window sites allows the preferential CO2 adsorption over CH4.
4. Sections 2.1 and 2.2:
a. Please include the Dubinin-Astakhov and Clausius-Clapeyron equation in the description of the methods.
5. Section 3.1:
a. Please include description of the parent natural ERI zeolite, of the preparation of the different ion-exchanged zeolites, and the main physico-chemical properties of all the zeolites studied, also from the starting material (chemical composition, e.g., cation content, Si/Al ratio, textural properties such as BET, external surface area). I believe they are necessary in order to be able to follow the discussion of the obtained adsorption capacities. In fact, at the beginning of section 3.2 (line 188) the authors remark that zeolites with different Si/Al ratios and different cations are being compared but give no further details.
b. Why does the micropore volume of the erionite zeolites increases after ion exchange with Na, Mg or Ca? Which are the cations present in the non-modified natural zeolite?
6. Section 3.2:
a. Figures 5 to 8 should be improved for facilitating comparison among the different zeolites studied: same symbols and colors should be used for the same temperatures in all plots. Moreover, using the same scales of the “y” axis would also make direct comparison easier. Why is the final pressure different for the different samples? Please explain. Why are the experiments performed in that range of pressures and temperatures? Please explain, as it is not obvious for readers not experts in the field of gas adsorption/separation. Why is the isosteric heat of adsorption calculated? What information does it give? How is it calculated? I have not found its description in section 2.
Conclusions are not clear enough. Which should be the selection criteria applied to erionite zeolites applied to H2 and CH4 capture, adsorption capacity or isosteric heat of adsorption?
Author Response
Nanomaterials-395938
Dear reviewer 1, we appreciate the comments and suggestions made to the manuscript "Critical admission temperature of H2 and CH4 in nanopores of exchanged ERI zeolites".
The revisions in the manuscript are marked up (in colored text) in order to make easily readable.
The detailed description of the changes made to the manuscript is shown in the attached file.
Miguel Ángel Hernandez
Zeolites Research Department, ICUAP, BUAP

Reviewer 2 Report
The ms. reports the effect of critical admission temperature through narrow pores of natural ERI zeolites at low levels of coverage by means of adsorption of CH4 and H2 on pores in natural erionite exchanged with aqueous solutions of Na+, Mg+2, and Ca+2 salts at different concentrations, variable time and temperature of treatment.
Despite of providing new and potentially interesting results, in my opinion the ms. suffers of a significant problem.
The authors indicate that the pristine material was characterized in Hernandez M.A.; Pestryakov A.; Portillo R.; Salgado A.; Rojas F.; Rubio E.; Ruiz S.; Petranovskii V. CO2 sequestration by natural zeolite for greenhouse effect control. Procedia Chemistry 2015, 15, 33-41. DOI:
10.1016/j.proche.2015.10.006. Available online:
https://www.sciencedirect.com/science/article/pii/S1876619615000856
However, despite the claim, that paper does not show any diffraction pattern clearly indicating the mineralogical composition of the raw material and of the exchanged ones.
This is a relevant point as the SEM-EDS analyses revealed the occurrence of large amounts of Fe (from ca. 1 to ca. 6 wt.%) that recent literature (see the various papers of Ballirano's research group) have been unequivocally attributed to clays and iron nano- oxides/hydroxides/sulfates... Moreover, the K content of the various samples is significantly << 2 apfu (atoms per formula unit) indicating problems in the chemical data (see same references that indicate such parameter as crucial for evaluating the quality of the chemical data). As a result, the calculated E% of the various samples is ca. -20%, which is further proof of unreliable data (should be < 10%).
Therefore, there are significant concerns about the real mineralogical composition of the samples used in the present ms. and I suggest to provide their detailed crystal chemical characterization, producing new SEM-EDS data, checking the well known critical chemical parameters and providing XRPD data testifying the phases identification, before resubmission of a revised version. In fact, chemical variability seems to suggest the occurrence of different proportions of ERI in the natural and cation-exchanged samples.
Author Response
Manuscript ID
Nanomaterials-395938
Dear reviewer 2, we appreciate the comments and suggestions made to the manuscript Nanomaterials-395938 "Critical admission temperature of H2 and CH4 in nanopores of exchanged ERI zeolites".
The revisions in the manuscript are marked up (in colored text) in order to make easily readable.
The detailed description of the changes made to the manuscript is shown in the attached file.
Dr. Miguel Angel Hernández
Zeolites Research Department, ICUAP, BUAP

Round 2
Reviewer 1 Report
General comments.
The authors have followed the suggestions made by the reviewers, and this second version of the manuscript is now more complete and easier to understand for researchers not so familiar with adsorption. Still, English usage should be revised, and I believe the manuscript could be improved further by correlating more thoroughly the characterization results (micropore volumes, TG-DTA results) with the final adsorption capacity of the different samples.
After some minor modifications the manuscript should be ready for its publication in Nanomaterials.
The details are given below.
Specific comments.
1. English usage should be revised
a. A recent revision on small pore zeolites with application in gas separation: Moliner et al., Chem. Mater. 2014, 26, 246−258
b. Shang et al., J. Am. Chem. Soc. 2012, 134, 19246. Use of cation exchanged (Na, K, Cs) CHA for CO2/CH4 separations. The paper shows how the different ability of gas molecules to induce cation deviation from their 8-ring window sites allows the preferential CO2 adsorption over CH4.
2. Section 3:
a. Description of the parent natural ERI zeolite is now included. When comparing composition and textural properties of this starting material with those of the modified zeolites I do not see significant changes in Si/Al ratio. What is clear is that the largest BET surface area is obtained after Na exchange, and that, when compared to the rest of the cations, this treatment is the most effective for K removal. Can the authors comment on this. Which is the location of the K ions? Is it the S3 site, as for Na?
b. I agree with the authors that part of the increase observed in the surface area of the ion exchanged samples can be due to the removal of amorphous debris. However, the size of the cation and the amount of cations exchanged will also play an important role in the remaining free micropore volume.
c. Which is the influence of the micropore volume in the adsorption capacity? I think it is important to separate the possible influence of a larger pore volume from other factors such as pore opening/closure due to cations location.
d. The Mg content of the Mg-exchanged erionite increases from 0.9 in the starting material to 1.43 after three ion exchanges. However, Fe content is doubled for samples ERIMg1 and ERIMg2 as compared to the natural zeolite. May the Fe content also play a role in the final adsorption properties? The same occurs for the Ca exchanged samples.
e. The TG-DTA study presented is complete and clearly explained. However, I miss a direct and explicit correlation of the information obtained in this section with the adsorption results of the final samples.

Author Response
Dear reviewer 1, we have corrected the manuscript according to the suggested changes.
We are very grateful for the comments and suggestions made to improve the manuscript.
a. Lines 160-170. Description of cationic sites of natural erionite in addition of the extraframework cation mobility in exchange samples was discussed (Figure 1, Table 2).
b. Lines 192-204. The relationship between the size of the cations, their position and the increase between the micropore volumes have been included.
c. We agree that the micropore volume plays a fundamental role in the adsorption processes as well as the other properties of texture, however these properties depend on the position of the cations and at the same time these have a temperature dependence which causes the migration of them in different positions. Therefore, if we evaluate the evaluation of the adsorption capacity at different temperatures then this value is variable. We improve the explanation of this dependence throughout the manuscript.
d. Lines 171-178. A new elemental composition analysis was performed by means of X-Ray Fluorescence, in which, although the Fe increase mentioned in the samples is no longer revealed, there is a considerable percentage of amorphous Fe2O3 as impurity of the sample, this presence of Fe in the samples does not affect the ability of adsorption to be an impurity and not be part of the structure of zeolites.
e. Lines 264-261. The discussion of the adsorption capacity correlated with the TGA analyzes was added.
Best Regards

Reviewer 2 Report
The ms. has been previously evaluated. The authors have modified the ms. with respect to the first submission. As a general comment, many of the Figures seem to be duplicated (see for example Fig. 1).
Detailed comments:
line 8: please write as [Al9Si27O72]
lines 86-87: cationic sites are generally labelled as Ca1, Ca2, Ca3, K1 AND K2 instead of S1-S4. In fact, there is a further cation site (K2) located near the walls of the erionite cavity which is occupied in the case of a chemical formula having K > 2 apfu (see for example Ballirano et al. 2009, American Mineralogist, 94, 1262-1270).
lines 105-106: X-ray powder diffraction (XRPD);
line 162 and 174: XRPD instead of XRD;
line 162 and following: the low quality pattern, apparently background subtracted, does not show reflections that can be assigned to further phases. However, as Table 2 indicates a relevant Fe content (this fact has been signaled in the first round of revision) and erionite, chabazite and quartz are nominally iron-free phases, at least a 10 wt.% of the sample is still unidentified (clays, iron oxy/hydroxides?).
As far as Table 2 is referred to, here I report the calculated chemical formulae of the various samples (in apfu) hypothesizing a content of ca. 30H2O per formula unit (18.5% H2O).
| ERINat | ERINa1 | ERINa2 | ERINa3 | ERICa1 | ERICa2 | ERICa3 | ERIMg1 | ERIMg2 | ERIMg3 | Avg | sd | |
| Na | 2,92 | 3,81 | 4,25 | 3,78 | 2,55 | 1,89 | 1,48 | 3,12 | 3,06 | 2,64 | 2,96 | 0,86 |
| Mg | 1,08 | 1,13 | 1,02 | 1,02 | 1,27 | 1,27 | 1,25 | 1,52 | 1,52 | 1,70 | 1,27 | 0,23 |
| Al | 7,43 | 7,72 | 8,19 | 7,82 | 7,57 | 7,69 | 7,77 | 7,72 | 7,61 | 7,45 | 7,70 | 0,22 |
| Si | 28,57 | 28,28 | 27,81 | 28,18 | 28,43 | 28,31 | 28,23 | 28,28 | 28,39 | 28,55 | 28,30 | 0,22 |
| K | 1,87 | 1,23 | 1,06 | 1,13 | 1,71 | 1,67 | 1,62 | 1,66 | 1,60 | 1,69 | 1,52 | 0,28 |
| Ca | 0,46 | 0,70 | 0,84 | 0,91 | 0,96 | 1,34 | 1,70 | 0,64 | 0,56 | 0,53 | 0,86 | 0,39 |
| Fe | 0,48 | 0,93 | 0,22 | 0,24 | 1,13 | 1,09 | 1,07 | 1,00 | 1,10 | 0,87 | 0,81 | 0,36 |
| H2O | 30,01 | 30,55 | 29,97 | 29,94 | 30,80 | 30,73 | 30,76 | 30,75 | 30,74 | 30,52 | 30,48 | 0,36 |
| O | 72,70 | 73,42 | 72,64 | 72,72 | 73,70 | 73,64 | 73,68 | 73,68 | 73,71 | 73,55 | 73,34 | 0,46 |
| E% | -15,9 | -26,9 | -13,5 | -15,6 | -31,0 | -29,8 | -30,2 | -30,4 | -31,0 | -29,4 | -25,8 | 7,3 |
As can be seen:
a) the K content is always << 2 apfu (average 1.5(3) apfu); 2 apfu are required to fill the 2 cancrinite cages of the framework.
b) the average content of Fe is of 0.8(4) apfu, with the exception of Na-exchanged samples, and this value increases upon exchange with respect to the pristine material.
c) the |E%| is >> 10 (average 26%).
All those clues indicate unreliable chemical data.
lines 177-178: Erionite occurs as bundles of parallel fibrils [32] with a diameter....
lines 218-223: how was this observed? No strucural data are reported. If this is a speculation, as such should be presented. There is a good number of recent works reporting the structure of exchanged samples of erionite indicating where different cation species are preferentially allocated.
line 233: observed instead of observe;
line 257: Figure 5a: 5 of 9 TG curves show weight increase during heating! Please explain!! If the mineralogical composition of your samples is only erionite + chabazite + quartz this is a non-sense. This is further proof that something is still missing.
line 313: unclear the meaning of S2-S3/S3-S4. Possibly, S2-S3-S4?
line 330: same as above;
Author Response
Dear reviewer 2, we have corrected the manuscript according to the suggested changes. We appreciate the comments and suggestions made to improve the manuscript "Critical admission temperature of H2 and CH4 in nanopores of exchanged ERI zeolites".
The answer to each observation can be found in the attached file. Additionally, the modifications were marked in colored text in the corrected manuscript.
Best Regards

Round 3
Reviewer 1 Report
The authors have followed the suggestions made by the reviewers, and this third version of the manuscript is now ready for publication in Nanomaterials. The location of the different cations is clearly explained, their size is explicitly discussed, and it is specified that Fe is present in a separate phase and not as compensation cation. Moreover, English usage has been improved.
I have only one brief suggestion on the Abstract. It will be more informative if the particular names of the samples, e.g., ERICa2 and ERINa3 in line 31, are substituted by a more general definition, such as Ca or Na exchanged ERI zeolite, for instance.